# VISION-LANGUAGE MODELS MEET METEOROLOGY: DEVELOPING MODELS FOR ANOMALIES ANALYSIS WITH HEATMAPS

## ABSTRACT

Real-time analysis and prediction of meteorological anomalies protect human lives and infrastructure. Traditional methods rely on numerical threshold setting and manual interpretation of weather heatmaps with Geographic Information Systems (GIS), which can be slow and error-prone. Our research redefines Meteorological Anomalies Analysis(MAA) by framing it as a Visual Question Answering (VQA) problem, thereby introducing a more precise and automated solution. Leveraging Vision-Language Models (VLM) to simultaneously process visual and textual data, we offer an effective aid to enhance the analysis process of weather heatmaps. Our initial assessment of general-purpose VLMs (e.g., GPT-4-Vision) on MAA revealed poor performance, characterized by low accuracy and frequent hallucinations due to *inadequate color differentiation* and *insufficient meteorological knowledge*. To address these challenges, we introduce **ClimateIQA**, the first meteorological VQA dataset, which includes 8,760 wind gust heatmaps and 254,040 question-answer pairs covering four question types, both generated from the latest climate reanalysis data. We also propose **Sparse Position and Outline Tracking (SPOT)**, an innovative technique that leverages OpenCV and K-Means clustering to capture and depict color contours in heatmaps, providing ClimateIQA with more accurate color spatial location information. Finally, we present **Climate-Zoo**, the first meteorological VLM collection, which adapts VLMs to meteorological applications using the ClimateIQA dataset. Experiment results demonstrate that models from Climate-Zoo substantially outperform state-of-the-art general VLMs, achieving an accuracy increase from 0% to over 90% in MAA verification.

## 1 INTRODUCTION

Meteorology is essential for disaster preparedness, agricultural planning, and climate resilience, as it encompasses the study of weather patterns, climate change, and severe weather events. One important task in meteorology is Meteorological Anomalies Analysis (MAA), which includes both anomalies detection and interpretation. MAA has become increasingly urgent due to the rise in these events driven by global warming. Traditionally, MAA relies on numerical threshold setting and manual analysis of weather anomaly heatmaps, which visualize data such as temperature, wind speed, and precipitation (Wunsch et al., 2022). Nonetheless, traditional methods using Geographic Information Systems (GIS) to analyze these heatmaps often require manual work, making the process time-consuming and error-prone.

In recent years, AI has advanced in processing large datasets and accelerating meteorological tasks. Machine learning models, such as Convolutional Neural Networks (CNNs), have been used for weather prediction and anomalies detection. Yet they mostly use numeric data and often yield low accuracy (Fang et al., 2021). Large Language Models (LLMs), such as ChatClimate (Vaghefi et al., 2023), ClimateGPT (Thulke et al., 2024), and ClimSight (Koldunov & Jung, 2024), excel at generating textual weather reports but struggle with interpreting visual data, which are an essential part in MAA.

Although General Vision-Language Models (VLMs) such as GPT-4-Vision (Achiam et al., 2023), Qwen-VL (Bai et al., 2023), and LLaVA 1.6 (Liu et al., 2024a) have broad abilities in visual

Figure 1: Comparative Analysis of Visual Chat and Reasoning Abilities in Meteorological Anomalies Analysis. Regions marked in yellow indicate strong breezes, red indicates hurricanes, and green indicates moderate breezes. In the conversation, hallucinations are marked in blue, refusal-to-answer responses are marked in red, and accurate responses are marked in green.

question answering, they struggle with MAA. Our case study (Figure 1) and initial assessment (Figure 2) demonstrate that these models often make mistakes when interpreting meteorological heatmaps. Three common issues observed are color misidentification, incorrect and irrelevant responses (hallucinations), and incomplete answers. A potential solution to these issues is to fine-tune VLMs using specialized meteorological data. However, prior meteorological datasets, for example, Extremeweather (Racah et al., 2017) and ClimSim (Yu et al., 2024), primarily focus on numeric analysis of anomalies detection. The lack of relevant guidelines and prior work leaves researchers with little guidance on creating effective images and Question-Answering pairs for MAA. To this end, our research uses a novel approach to identify issues underlying the poor performance of VLMs on MAA and propose potential solutions. The contributions of this work are:

1. We identify a set of issues and corresponding solutions for improving VLMs performance in heat map-based MAA. These findings can serve as a baseline framework for future efforts in this field.

2. We propose a novel method, Sparse Position and Outline Tracking (SPOT), to detect anomalies and obtain spatial information on colored regions in meteorological heatmaps. SPOT uses K-Means (Krishna & Murty, 1999) to obtain color representations (Figure 3). Experiments show that color spatial location obtained via SPOT has a 100% accuracy.

3. We release the first meteorological VQA dataset, ClimateIQA. It consists of 8,760 high-resolution images and 254k instruction samples. Compiled from ERA5 (Hersbach et al., 2020) reanalysis data, data processed by SPOT, geography knowledge bases (Programme, 2019; Institute, 2018), and the Beaufort Scale (Monmonier, 2005), ClimateIQA is designed to train VLMs to identify anomalies, as well as interpret and describe meteorological heatmaps.

4. We introduce Climate-Zoo, the first collection of meteorological VLMs built upon state-of-the-art VLMs (e.g., Qwen-VL-Chat (Bai et al., 2023), Llava 1.6 (Liu et al., 2024a), and Yi-VL (Young et al., 2024)). Climate-Zoo substantially outperforms existing models on meteorological heatmap anomalies interpretation and can effectively localize areas of anomalies, setting a new benchmark for meteorological AI tools.

## 2 RELATED WORK

### 2.1 VISION LANGUAGE MODELS AND VISUAL QUESTION ANSWERING

The integration of visual and textual data has led to the development of advanced VLMs, which typically build upon the capabilities of text-only LLMs, such as GPT-4 (Achiam et al., 2023), LLaMA (Touvron et al., 2023), Gemini (Team et al., 2023), and Claude (Anthropic, 2024). Notable developments in VLMs include GPT-4-vision (Achiam et al., 2023), Qwen-VL (Bai et al., 2023), and LLaVA (Li et al., 2024), which have substantially enhanced the efficiency of VQA tasks. These tasks require models to comprehend and respond to information and questions in both visual and textual formats.

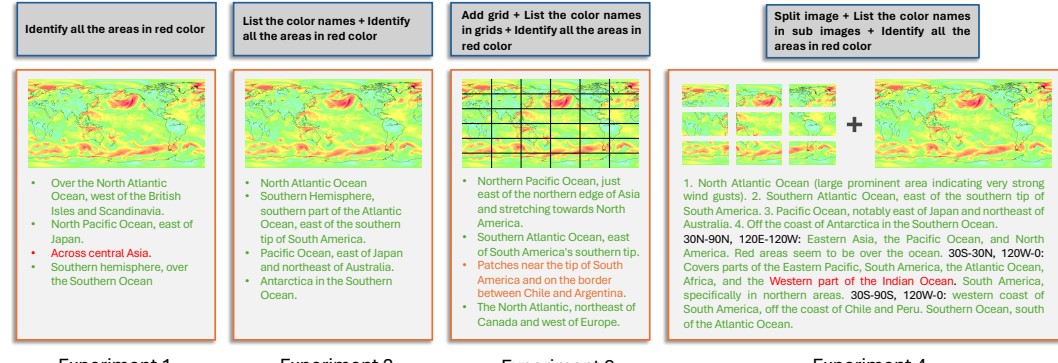

Figure 2: Result of the Initial Assessment via Prompt-Tuning GPT-4-Vision. Sentences in red mark inaccurate responses, sentences in orange and black mean surprising findings (patches and geography coordinates), and sentences in green mark accurate answers.

To enhance model performance in VQA, researchers have adopted advanced methods for visual feature extraction (Zheng et al., 2023), developed robust model architectures (Liu et al., 2024a), and explored innovative learning paradigms (Chen et al., 2024). Despite these advancements, VQA tasks continue to face challenges, such as the occurrence of hallucinations (Bai et al., 2024), often stemming from issues like data quality and visual uncertainty (Leng et al., 2023). Addressing these issues highlights the critical need for high-quality datasets and effective strategies to mitigate challenges in VQA tasks.

### 2.2 AI FOR METEOROLOGY

The integration of AI in meteorology has seen many applications, such as employing AI for long-term weather prediction (Lam et al., 2022), typhoon trajectory forecasting (Bi et al., 2022), and weather classification (Dalal et al., 2023). Models like Pangu-weather Bi et al. (2023), Fengwu Chen et al. (2023), and NeuralGCM Kochkov et al. (2024) are outstanding. The advent of LLMs like ClimSight (Koldunov & Jung, 2024), ChatClimate (Vaghefi et al., 2023), Arabic Mini-ClimateGPT (Mullappilly et al., 2023), and ClimateGPT (Thulke et al., 2024) has broadened the scope of textual data processing in meteorology. These models have been instrumental in assimilating general meteorological knowledge related to climates, answering common queries, and offering insights. However, these models predominantly rely on textual data. This becomes particularly limiting when addressing complex challenges such as the analysis of anomalies distributions in heatmap, where textual data alone proves inadequate and prone to inaccuracies, often leading to serious hallucinations (Bulian et al., 2023). Meteorologists often need to interpret data from satellite images (Liu et al., 2024b), radar (Guastavino et al., 2022), heatmaps (Lee et al., 2024), and isobaric maps (Xu et al., 2024) to make accurate assessments. Nonetheless, there remains a lack of VLMs capable of interpreting such visual meteorological data.

## 3 INITIAL ASSESSMENT OF GPT-4-VISION

Among various VLMs, GPT-4-Vision (Achiam et al., 2023) has demonstrated exceptional capabilities in understanding and generating visual and textual content (Singh et al.). We began with an in-depth evaluation of its ability to identify and localize red regions in heatmap images, indicating areas like high wind speed, temperatures, or significant weather metrics, aiming to pinpoint areas for enhancement based on its limitations. Four experiments were designed for this assessment (Figure 2):

1. Direct Red Region Identification: We tested the VLM's ability to identify red regions directly, without guidance, to evaluate its color perception and localization capabilities.

2. Two-Step Color Identification: After observing potential color confusion in the first experiment, the process was altered to first list all colors in the image, then identify red regions specifically, improving the accuracy of color recognition.

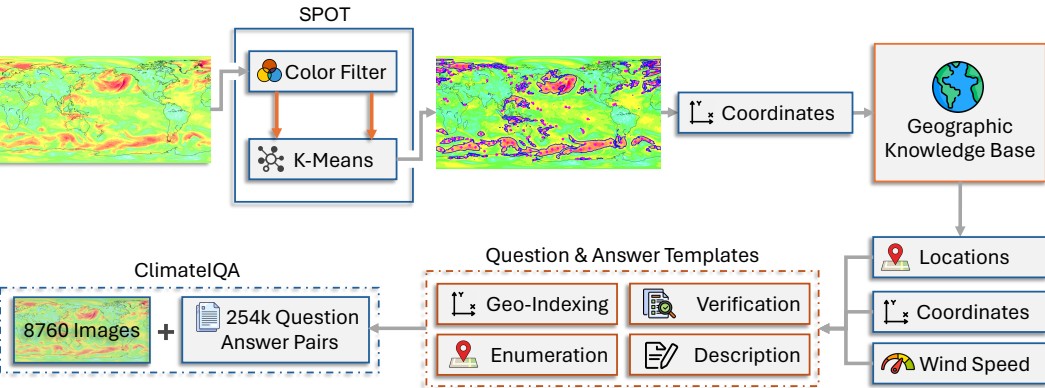

Figure 3: Dataset Creation Process. Images were processed using SPOT to extract color contours (marked in blue) and representative point coordinates (marked in purple), such as (-40, 65). The extracted data were integrated into geographic knowledge bases to retrieve location-specific information. These data, including location, coordinates, and wind speed, were then input into predefined question-and-answer templates, resulting in the generation of 254,040 question-answer pairs. The final dataset, ClimateIQA, pairs these QA pairs with 8,760 images, enabling comprehensive climate-related visual question answering.

3. Grid-Based Color Identification: To capture fine-grained details, we implemented a grid-based method, dividing images into a $6 \times 6$ grid, each with geographic details. The model identified all colors which are present in the cell and then located the red-colored regions, evaluating the model's ability to capture local color information and its impact on localization accuracy.

4. Image Segmentation and Combined Analysis: Employing image segmentation via the PIL toolkit (Umesh, 2012), we divided the input image into sub-images and tasked the VLM with analyzing both the overall and segmented images. The results were then combined for a comprehensive interpretation, aiming to improve the completeness and accuracy of the model's responses.

The results varied across experiments. In Experiment 1, GPT-4-Vision struggled with direct identification of red regions, inaccurately marking locations such as "Across central Asia". Experiment 2 showed improvement with correct identifications, though responses were incomplete and the recall rate was just 5%. Experiment 3, the grid-based approach, better-captured details like patches but had inconsistent performance across different images, with an average accuracy of 7%. Experiment 4 utilized a segmented and combined analysis approach, yielding the most accurate results among our trials. The model successfully identified sub-image colors and provided more detailed interpretations, including specific geographic coordinates and thorough annotations. Despite these improvements, the responses were still incomplete, with an average recall rate of only 12%. Additionally, similar to Experiment 3, erroneous results occurred when segmented image analysis led to incorrect color judgments. The increased number of generated answers correlated with a higher error rate, highlighting a critical area for further enhancement.

Overall, these experiments highlight areas for improvement in VLMs, particularly in addressing color confusion and enhancing geographical knowledge. Despite improvements in image segmentation and combined analysis, further refinements are necessary for more accurate and reliable performance in identifying and localizing colored regions in images.

# 4 CLIMATEIQA DATASET CREATION

As shown in Figure 3, we ensured the reliability of the image sources and developed a new method for accurately extracting color and position information from the images.

## 4.1 DATA COLLECTION AND PRE-PROCESSING

Our meteorological data were derived from the ERA5 hourly dataset on single levels. ERA5, produced by the European Centre for Medium-Range Weather Forecasts (ECMWF) (Hersbach et al., 2020), provides comprehensive global climate and weather records since 1940. This dataset is created using data assimilation techniques that integrate model outputs with observational data, enhancing accuracy by reconciling forecasts with new observations every 12 hours. For this study, we selected hourly wind gust data for 2023. Due to the complexity of wind gust heatmaps, which contains two times colors than precipitation or temperature heatmaps, wind gust data is especially representative for identifying anomalies. The method developed with wind gust data can be conveniently extended in the future to analyze less complex heatmaps, such as heatmaps for heatwaves, droughts, and heavy precipitation.

To classify wind speeds, we employed the Beaufort Scale (Monmonier, 2005), a widely recognized system that quantifies wind speed by observing its impact at sea or on land. The Beaufort Scale categorizes wind speeds from 0 to 12, with each level associated with a specific wind speed range and descriptive physical conditions. In our analysis, each wind force level is represented by a unique color gradient, starting from white for the lowest wind speeds and progressing through light cyan, aquamarine, light green, light lime green, light lemon yellow, light yellow, peach, light coral, salmon, deep pink, and dark magenta, culminating at dark purple for the highest wind speeds. In meteorological research, Beaufort scale level 8 (20.8 m/s) is commonly used to demarcate extreme weather conditions Radinović & Ćurić (2014); Weaver et al. (2021). Therefore, we marked colors after "peach" as anomalies in the heatmaps. To facilitate the geographical localization of anomalies, we superimposed a world map onto each heatmap.

To extract and compile color information from meteorological heatmap images, we developed a method called "SPOT" (Sparse Position and Outline Tracking). Here are the process:

**1. Color Segmentation:** Initially, our SPOT method extracts contours from heatmaps based on four primary colors: red, yellow, white, and green, using OpenCV as the color filter (Culjak et al., 2012). We obtain the contour coordinates of each color region to address the issue of irregular shapes often encountered in heatmaps. This process is iterated twice to ensure accuracy, selecting the best segmentation result to mitigate errors.

**2. Representative Point Selection:** Given the large volume of contour coordinate data, we represent each color region's geographical location and distribution shape using a minimal set of points. We start by determining the number of points needed based on the area of each color region within the image. To tackle the challenge of representing irregularly shaped color regions with a few coordinates, we applied the K-Means clustering algorithm to compute the centroid coordinates for each region. We set the random state to 0 to ensure reproducibility and eliminate randomness. The number of clusters (k) is determined by the area of the color regions: Less than 1% of the total area: 1 point. 1%-5% of the total area: 3 points. 5%-10% of the total area: 5 points. More than 10% of the total area: 10 points.

**3. Filtering Outliers:** We implemented a rule-based function to ensure all points fall within their respective color regions. Any points found outside these regions are automatically excluded and replaced with new points from the nearest valid contour. In a processed heatmap containing 5,448 points, approximately 122 points may fall outside the contour, resulting in an efficiency rate of about 97.7%. Our method reroutes these outlier points to maintain the robustness and accuracy of the model. As shown in Figure 3 of our paper, each purple dot precisely represents the spatial location and shape of the corresponding color region.

As illustrated in Figure 3, each purple dot represents the position and shape of its corresponding color region. With the help of the SPOT method, we can correctly identify the spatial location of different color regions with 100% accuracy. The pseudo-code of SPOT is in Appendix 1.

## 4.2 CREATING INSTRUCTION-TUNING DATA

After identifying the representative points for each color block using SPOT, we indexed the corresponding geographical names of these points coordinates using two geographic databases: the IHO Sea Areas (Institute, 2018) and the World Bank-approved Administrative Boundaries (Programme,

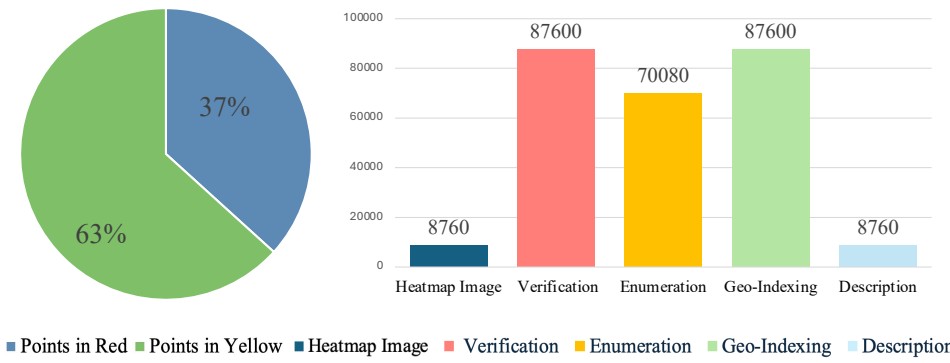

Figure 4: Distribution of red and yellow point coordinate data collected by SPOT (left) and the final ClimateIQA dataset (right).

2019). The IHO Sea Areas database delineates the boundaries of the world's major oceans and seas, while the World Bank-approved administrative boundaries database includes international borders, disputed areas, coastlines, lakes, and a usage guide.

We then designed templates (Table 6) for question and answer generation, ensuring that each question and answer pair would be generated based on the templates and could be substantiated by the data. We filled in the blanks of templates with essential information such as specific locations, coordinates to systematically generate the corresponding question and answer pairs, forming the instruction-tuning data. As shwon in Table 6, we divided all the instruction-tuning data into four different types: (1) Verification questions that determine whether a location in the heatmap has anomalies ; (2) Enumeration questions that list all the places that have anomalies in the whole heatmap; (3) Geo-Indexing questions that provide the coordinates of a anomalies in the heatmap; and (4) Description questions that provide a detailed interpretation of anomalies for the given image.

The development of these four tasks is based on the limitations identified during our initial assessment experiments with VLM in session three. Specifically, we identified that the VLM lacked sufficient geographic and meteorological knowledge, leading to incorrect answers, inaccurate color localization, and incomplete responses. To address these issues, we constructed the following four tasks, each targeting a specific area of improvement: Verification Questions aims to enhance the model's accuracy in identifying anomalies, which is critical for timely and precise weather forecasting. Geo-Indexing Questions focuses on improving the model's capability to accurately locate colors within images, which is essential for correct geographical referencing and the interpretation of meteorological data. Enumeration Question is designed to enhance the completeness of the model's responses, ensuring that all relevant aspects of a query are adequately addressed. Description Questions involves generating comprehensive reports, which are vital for detailed meteorological analysis and communication of weather-related findings. Each of these tasks is meticulously crafted to address specific weaknesses in VLM, thereby improving its overall performance in anomalies recognition and analysis.

## 4.3 DATASET STATISTICS

Our approach produced 8,760 high-resolution heatmaps, each measuring $3510 \times 1755$ pixels. These images provide detailed visual representations of global wind patterns. An example of instruction-tuning data is shown in Figure 6. With geographical names fixed (Figure 4), the question type distribution is as follows: Verification Questions (34.5%), List Questions (27.6%), Geo-Indexing Questions (34.5%), and Description Questions (3.4%). We focused on localizing anomalies, collecting points in red (wind speeds exceeding 20.8 m/s) and yellow (wind speeds between 10.8-20.7 m/s). Of the data collected by SPOT, 37% of the points are red, and 63% are yellow.

This process yielded a large-scale instruction-tuning dataset of 254,040 data points. We further split it into training, validation, and testing sets in a 7:1:2 ratio, with 177,828 instruction samples for training, 25,404 for validation, and 50,808 for testing. We designate 203k, including both training and validation data, as fine-tuning dataset.

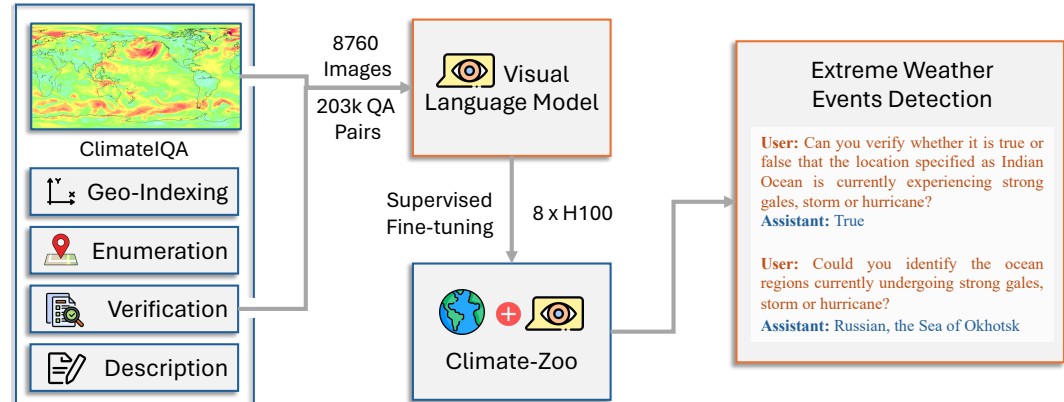

Figure 5: Process Supervised Fine-tuning Climate-Zoo.

Meanwhile, as employing a less frequent sampling strategy would mitigate redundancy and enrich the dataset with more unique information, we also create a ClimateIQA-daily using daily wind gust data, anchored at 00 UTC. ClimateIQA-daily contains only 365 images and has the same question type ratio distribution as ClimateIQA.

## 5 CLIMATE-ZOO: ADAPTING VLMS TO METEOROLOGY

This section details our methods to enhance the performance of VLMs on MAA through prompt-tuning and supervised fine-tuning with ClimateIQA.

**Base models**   Based on model performance on VLM benchmarks (Goyal et al., 2017; Lu et al., 2022), we selected three state-of-the-art VLMs as our base models for improvement: Llava-v1.6-mistral-7b (Liu et al., 2024a), Qwen-VL-Chat (Bai et al., 2023), and Yi-VL-6B (Young et al., 2024). Llava-v1.6 excels in multimodal understanding, Qwen-VL-Chat in visual dialog tasks, and Yi-VL-6B in visual reasoning.

**Supervised fine-tuning**

Supervised fine-tuning involves instructing a pre-trained model to improve its performance on a specific task by providing task-specific information. As depicted in Figure 5, we used 70% of the data for training, 10% for validation, and 20% for testing. Our strategy included freezing the weights of the visual encoder and employing a unified encoder layer to reduce computational costs and mitigate overfitting risks. This approach is supported by (Khattak et al., 2023), who demonstrated that pre-trained visual encoders are proficient at extracting meaningful features. Additionally, the size of the training dataset significantly impacts fine-tuning effectiveness. We conducted experiments with different dataset sizes (10k, 50k, 100k, and 203k) to evaluate the effects on fine-tuning performance and ultimately selected the best-performing model.

**Training details**   We conducted full-parameter training on three prominent large-scale VLMs: Llava-v1.6 (7B parameters), Qwen-VL-Chat (7B parameters), and Yi-VL-6B (6B parameters), utilizing the Swift toolkit for its efficiency and flexibility. In addition to full-parameter tuning, we also fine-tune these models with Low-Rank Adaptation (LoRA) layers to further improve their adaptability. LoRA introduces a low-rank decomposition of the model's weight matrices, allowing efficient adaptation to new tasks with minimal additional parameters. By setting the LoRA rank to 8, with an alpha value of 32 and a dropout probability of 0.05, we balance adaptation capacity and computational efficiency.

To expedite the training process, we employed 8 H100 80G GPUs and utilized Distributed Data Parallel along with DeepSpeed. The batch size was set to 1, and the learning rate was 0.0001 (1e-4). The entire experiment was conducted for a single epoch, spanning a total duration of 22 days.

Table 1: Result of Supervised Fine-tuning

| | Model | F1 Score ↑ | Element Match Score ↑ | Haversine Distance $10^3$ ↓ | BLEU ↑ | ROUGE ↑ | GPT4-Score (Similarity) ↑ | GPT4-Score (Total) ↑ |
|---|---|---|---|---|---|---|---|---|
| | Qwen-VL-Chat | 0 | -1 | 6.928 | 0 | 0.08 | 1 | 1.455 |
| | Yi-VL-6B | 0 | -1 | 6.718 | 0.004 | 0.052 | 2 | 3.035 |
| Baseline Model | Llava-v1.6-mistral-7b | 0 | -1 | 8.566 | 0 | 0.041 | 1.988 | 3.100 |
| | GPT-4-Vision | 0 | -1 | - | 0 | 0 | 2.012 | 3.186 |
| | Qwen-VL-Chat LoRa | 0.909 | -0.930 | 1.894 | 0.819 | 0.732 | **4.861** | 4.431 |
| | Yi-VL-6B LoRa | 0.905 | -0.934 | **1.887** | 0.007 | 0.055 | 1.850 | 2.868 |
| Climate-Zoo LoRa | Llava-v1.6-mistral-7b LoRa | 0.910 | -0.822 | 1.905 | 0.821 | 0.731 | 4.731 | 4.373 |
| | Qwen-VL-Chat | 0.910 | **-0.012** | 1.928 | 0.818 | 0.722 | 4.829 | **4.486** |
| | Yi-VL-6B | **0.912** | -0.122 | 1.933 | 0.815 | 0.728 | 4.741 | 4.360 |
| Climate-Zoo | Llava-v1.6-mistral-7b | 0.897 | -0.483 | 1.935 | **0.823** | **0.747** | 4.806 | 4.444 |

# 6 EVALUATION

## 6.1 METRICS

We designed a comprehensive framework to assess the performance of the models across four categories of questions. Our goal was to understand the model's adaptability and accuracy across varied tasks. Below, we detail our methodological approach and assessment metrics.

**F1 Score**   For Verification Questions, we evaluate the model's ability to judge the correctness of statements using recall, precision, and F1 score (the harmonic mean of precision and recall).

**Element Match Score**   For Enumeration Questions, we compute a match score (MS) between the ground truth ($x$) and model-generated answer ($y$). This involves comparing the sets ($x$) and ($y$) formatted as ["New York", "High Seas of the North Atlantic Ocean", "Canary Islands"], representing ground truth and model output, respectively. The match score calculation involves determining correct matches via the set intersection size (common elements in both sets ($x$) and ($y$)) and incorrect matches via the symmetric differences (elements present in one set but not in the other). In cases where both sets ($x$) and ($y$) are empty (union size of zero), the match score is defined to handle division by zero and set to zero. Otherwise, the match score ranges between -1 and 1, where a score closer to 1 indicates more accurate and complete answers with fewer hallucinations (incorrect items), and a score closer to -1 indicates poor performance with many hallucinations. The score is determined by the formula:

$$MS = \begin{cases} 0, & \text{if } |x \cup y| = 0 \\ \frac{|x \cap y| - (|x - y| + |y - x|)}{|x \cup y|}, & \text{otherwise} \end{cases} \quad (1)$$

**Haversine Distance**   For Geo-indexing Questions, which involve determining precise geographical coordinates, we utilized the Haversine distance formula. This metric accurately measures the distance between model-generated coordinates ($lat_m$, $lon_m$) and ground truth coordinates ($lat_g$, $lon_g$) by accounting for the Earth's curvature. The formula is as follows, where $r$ represents the Earth's radius:

$$d = 2r \cdot arcsin(\sqrt{sin^2(\frac{lat_m - lat_g}{2}) + cos(lat_m) \cdot cos(lat_g) \cdot sin^2(\frac{lon_m - lon_g}{2})}) \quad (2)$$

**BLEU, ROUGE and GPT-4 Scores**   For Description Questions, we employ average BLEU Papineni et al. (2002) and ROUGE Lin (2004) and GPT-4 scores Cao et al. (2024). BLEU-1 and BLEU-2 measure linguistic accuracy by comparing n-grams between the generated and ground truth descriptions. ROUGE-1, ROUGE-2, and ROUGE-L assess the overlap of n-grams, word sequences, and word pairs, offering insights into the comprehensiveness and relevance of the generated descriptions. Additionally, GPT-4 evaluates the overall quality and similarity of model-generated descriptions to ground truth answers on a five-point Likert scale. The prompt is shown in Appendix 7.

## 6.2 RESULTS AND ANALYSIS

**Supervised fine-tuning**   Table 1 illustrates the outcomes of our experiments, highlighting that Climate-Zow models outperform all baseline models across various metrics. Specifically, for verification and enumeration questions, the baseline models were unable to provide answers, which is

Table 2: Result of ablation study

| Climate-Zoo Model | Dataset | F1 Score | Element Match Score | Haversine Distance $10^3$ | BLEU | ROUGE | GPT4-Score (Similarity) | GPT4-Score (Total) |
|---|---|---|---|---|---|---|---|---|
| Yi-VL-6B | ClimateIQA-10k | 0.909 | -0.092 | **1.930** | **0.820** | **0.732** | **4.855** | **4.594** |
| | ClimateIQA-50k | 0.905 | -0.070 | 1.943 | **0.820** | 0.728 | 4.687 | 4.422 |
| | ClimateIQA-100k | **0.912** | **-0.048** | 1.932 | 0.814 | 0.718 | 4.834 | 4.345 |
| | ClimateIQA-203k | **0.912** | -0.122 | 1.933 | 0.815 | 0.728 | 4.741 | 4.360 |
| Llava-v1.6-mistral-7b | ClimateIQA-10k | 0.820 | -0.913 | 6.335 | 0.611 | 0.624 | 4.631 | **4.508** |
| | ClimateIQA-50k | 0.825 | -0.903 | 1.945 | 0.820 | 0.748 | 4.787 | 4.489 |
| | ClimateIQA-100k | 0.820 | -0.532 | 1.972 | **0.825** | **0.750** | 4.662 | 4.394 |
| | ClimateIQA-203k | **0.897** | **-0.483** | **1.935** | 0.823 | 0.747 | **4.806** | 4.444 |

reflected in F1 scores of 0 and match scores of -1. In stark contrast, Climate-Zoo models demonstrated an impressive accuracy of around 90% in pinpointing regions with anomalies, with the highest element match score reaching -0.012, indicating minimal inaccuracies in the data provided. Nevertheless, Climate-Zoo models did yield slightly incomplete lists of affected areas.

In tasks like geo-indexing and description questions, where baseline models did manage to generate responses, they were often plagued by significant errors. On the other hand, Climate-Zoo models significantly outperformed these baseline counterparts by delivering more precise coordinates and more accurate, rich descriptions, achieving superior BLEU, ROUGE, and GPT-4 scores.

While LoRA fine-tuning generally reduces the need for computational resources and, in specific cases like geo-indexing, even outperforms full parameter tuning, it doesn't universally enhance performance across all models. Notably, the Yi-VL-6B LoRA model falls short in handling description questions, underperforming both the fully fine-tuned models and the baseline.

Within the diverse ensemble of the Climate-Zoo collection, each model demonstrates particular strengths. The Qwen-VL-Chat model shines in detecting anomalies within a heatmap and providing detailed, vibrant image narratives, achieving high GPT scores. Conversely, the Yi-VL-6B model stands out with the highest F1 score, showcasing its accuracy in confirming anomalies at pinpoint locations. Meanwhile, the Llava-v1.6-mistral-7b model excels in spatial accuracy and textual richness, as evidenced by its exceptional performance in Haversine Distance, BLEU, and ROUGE evaluations, making it adept at generating precise coordinates and detailed visual descriptions.

**Ablation study** Table 2 presents the results of an ablation study using the Climate-Zoo models (Llava-v1.6-mistral-7b and Yi-VL-6B) with full parameters. This study evaluates model performance across varying dataset sizes: 10k, 50k, 100k, and 203k samples. Our findings reveal that increased data volume does not always correlate with improved model performance, with variations observed both between models and across different question types. At the model level, the Yi-VL-6B model achieves excellent results with just 10k samples; increasing the dataset size beyond this point can actually degrade its performance. In contrast, the Llava-v1.6-mistral-7b model shows improved performance with larger datasets. At the question type level, verification and enumeration questions demonstrate better performance with larger training datasets, whereas geo-indexing and description questions exhibit more variability.

Overall, the impact of dataset size on model performance varies significantly among different models. The Yi-VL-6B model appears especially suitable for industrial applications, as it can achieve high effectiveness with smaller datasets and fewer computational resources. We have delved into the potential reasons behind the exceptional performance of the Yi-VL-6B model with the smallest dataset. Our hypothesis centers on the unique characteristics of the pre-training dataset used for Yi-VL-6B. Unlike other VLMs, the Yi-VL-6B model was pre-trained on an extensive dataset comprising 34 billion tokens sourced from encyclopedic texts, which inherently include a significant amount of meteorological and geographical content. This pre-training on domain-rich data likely endowed the model with a robust foundation in meteorological concepts and terminology. As a result, Yi-VL-6B is primed to assimilate new information in this domain with minimal fine-tuning, allowing it to achieve outstanding performance even with a limited dataset.

## 7 LIMITATIONS

Although our VLMs demonstrate potential in detecting anomalies, their accuracy, currently at 91%, can be further enhanced. A notable limitation is the model's difficulty in accurately identifying colors in heatmaps, largely due to training data comprising only fully intact heatmaps. Inspired by the chain-of-thought prompting technique, we propose dividing the original heatmap into nine sub-images to generate individual question-answer pairs for fine-tuning. This approach aims to improve the model's color localization skills, ultimately enhancing accuracy. Additionally, integrating VLMs with traditional threshold-based methods could create a robust hybrid system, combining interactive strengths with precise anomaly detection.

Meanwhile, our initial dataset, focused solely on wind gust data due to its complexity (13 colors), poses another limitation. Future research should incorporate additional weather factors like temperature and precipitation to build a more comprehensive dataset. This would enable more generalized, robust VLM training, improving performance across various anomalies detection tasks.

## 8 CONCLUSIONS

This study presents pioneering work in integrating VLMs for MAA, providing a robust framework for anomalies detection and interpretation. Through a series of carefully designed experiments, we identified key issues of current general-purpose VLMs and proposed a framework of solutions to improve their performance. We introduced the SPOT method for precise color localization, developed ClimateIQA, the first large-scale dataset in this field, and established Climate-Zoo, a collection of state-of-the-art VLMs adapted for meteorological applications. Models based on Qwen-VL-Chat and Llava-v1.6-mistral-7b showed exceptional performance across verification, enumeration, geo-indexing, and description tasks of MAA. This research pushes the boundaries of AI usage in meteorology and contributes a benchmark for heatmap-based MAA, guiding future research to dive further into this field.

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

## A    APPENDIX

### A.1    LISCENSE

The ClimateIQA dataset and Climate-Zoo will be publicly available and use the **CC BY 4.0 license**.

The IHO Sea Area and the World Bank-approved Administrative Boundaries datasets, used to create ClimateIQA, are licensed under **CC BY 4.0 license**.

The ERA5 dataset is available under a free, worldwide, non-exclusive, royalty-free, and perpetual license. According to this license, access to Copernicus Products is granted for any lawful purpose. Permissible uses include, but are not limited to, reproduction, distribution, public communication, adaptation, modification, and combination with other data and information.

### A.2    EXAMPLE OF QUESTION TYPES

The following Figure 6 shows the four question types in the ClimateIQA instruction tuning data.

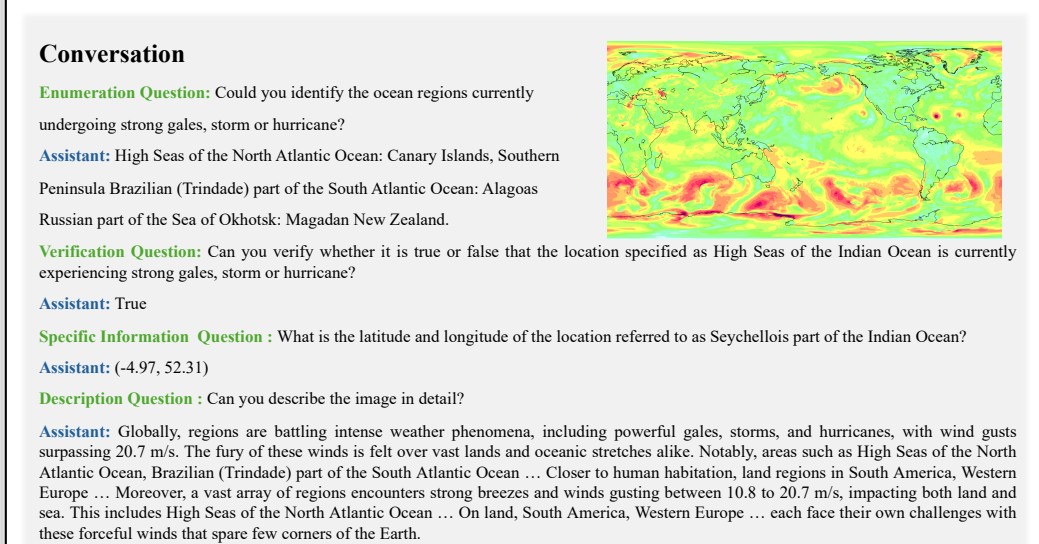

Figure 6: Example of Visual Question Answering Question Types in the Climate Domain.

A.3    ANSWER GENERATED IN PRELIMINARY STUDY

Table 3: An Example of Experiment One

| | |
|---|---|
| **Prompt** | This is a global wind gust heatmap. Could you please identify the areas are red color? |
| **Answer** | Based on the colors on this global wind gust map, the red areas, which typically indicate regions of higher wind gusts, appear in several locations. Here are the general areas where red coloration is present:
1. Over the North Atlantic Ocean, west of the British Isles and Scandinavia, indicating a zone of high wind gust activity possibly associated with a storm system. 2. In the North Pacific Ocean, east of Japan, which might be showing strong wind gusts also associated with a storm system. 3. Across central Asia, which could be indicative of strong winds blowing over the steppes or associated with a weather system in the region. 4. A region in the southern hemisphere, possibly over the Southern Ocean, indicating strong wind gusts which are common in those latitudes known as the roaring forties and furious fifties.
It's important to note that without a legend or scale, the exact wind speeds corresponding to the red color cannot be determined. |

Table 4: An Example of Experiment Two

| | |
|---|---|
| **Prompt** | This is a global wind gust heatmap. please list the color name in the image. Could you please identify the areas are red color? |
| **Answer** | The image shows different colors representing varying wind gust intensities across the globe. Here are the prominent colors visible in the image:
1. Green - varying shades indicating moderate wind gusts. 2. Yellow - indicating stronger wind gusts compared to green areas. 3. Orange - indicating even stronger wind gusts, higher than yellow areas. 4. Red - representing the strongest wind gusts visible on the map.
As for the red areas:
- There is a large, prominent red area over the North Atlantic Ocean, indicating very strong wind gusts there. - In the Southern Hemisphere, another visible red area is over the southern part of the Atlantic Ocean, east of the southern tip of South America. - There are also red areas in the Pacific Ocean, notably east of Japan and northeast of Australia. - Another notable red area is visible just off the coast of Antarctica in the Southern Ocean.
Keep in mind that identifying specific geographic locations just based on colors on a map can be challenging without specific map coordinates, country borders, or other geographical markers. |

Table 5: An Example of Experiment Three

| | |
|---|---|
| **Prompt** | This is a global wind gust heatmap. there are black grids in the image, from left to right is the longitude, there are 6 columns, each column stand for 0 60E,60E 120E,120E 180,180 120W, 120W 60W, 60W 0W. from bottom to top is the latitude, there are 6 rows, each row stands for 90S 60S, 60S 30S, 30S 0, 0 30N, 30N 60N, 60N 90N. Please list the color name in each grid. Could you please based on the color name in grids identify all the areas in red color? |
| **Answer** | Here are the predominant color names in each grid, starting from the leftmost column (0 60E) to the rightmost column (60W 0W) and from bottom to top (90S 60S to 60N 90N): |

1. 0 60E - 90S 60S: Light green and yellow - 60S 30S: Green with patches of red and yellow - 30S 0: Yellow with areas of green and red - 0 30N: Green, yellow, and small areas of red - 30N 60N: Green transitioning to yellow and red - 60N 90N: Yellow and red with patches of green

2. 60E 120E - 90S 60S: Light green and yellow - 60S 30S: Yellow with areas of red and green - 30S 0: Yellow and green with small patches of red - 0 30N: Green with some yellow - 30N 60N: Yellow and red with hints of green - 60N 90N: Green and yellow with small areas of red

3. 120E 180 - 90S 60S: Light green and yellow - 60S 30S: Green and yellow with red areas - 30S 0: Yellow with areas of green and small patches of red - 0 30N: Green and yellow with hints of red - 30N 60N: Green and yellow with red areas - 60N 90N: Yellow with patches of red

4. 180 120W - 90S 60S: Light green and yellow - 60S 30S: Light green - 30S 0: Green and yellow with areas of red - 0 30N: Green and yellow with small patches of red - 30N 60N: Green and yellow with areas of red - 60N 90N: Green and yellow

5. 120W 60W - 90S 60S: Light green and yellow - 60S 30S: Green with patches of red and yellow - 30S 0: Green and yellow - 0 30N: Green and yellow - 30N 60N: Yellow and green with red areas - 60N 90N: Green and yellow with patches of red

6. 60W 0W - 90S 60S: Light green and yellow - 60S 30S: Yellow and green with areas of red - 30S 0: Yellow with patches of green and red - 0 30N: Green and yellow with small areas of red - 30N 60N: Yellow and green with red areas - 60N 90N: Green and yellow with areas of red

The colors on a heatmap such as this generally range from cooler colors (like greens and blues) representing lower values to warmer colors (like yellows, oranges, and reds) representing higher values. Since there are no distinct boundaries, naming a single color for each grid would not be accurate.

As for the red areas on the map, they indicate regions of high wind gusts. Describing their locations with precision is difficult because of the blending colors and the lack of borders on the map, but I can provide a general idea of where the reddest areas are:

1. A very intense red area is located in the Northern Pacific Ocean, just east of the northern edge of Asia and stretching towards North America. 2. Another significant red area is in the southern Atlantic Ocean, east of South America's southern tip. 3. There are patches of red near the tip of South America and on the border between Chile and Argentina. 4. The North Atlantic, northeast of Canada and west of Europe, also contains some red areas.

## A.4 PSEUDO CODE OF SPOT

---

**Algorithm 1** SPOT: Sparse Position and Outline Tracking

---

**Input:** Image path $img\_path$, color name $color\_name$
**Output:** List of latitude and longitude coordinates
1: Initialize $GetCoordinate$ with $img\_path$
2: **procedure** GETCOLORBOUNDARIES($image, color\_name$)
3:      Convert $image$ to HSV color space
4:      Generate mask based on color range for $color\_name$
5:      Find contours in the mask
6:      **return** contours, mask
7: **end procedure**
8: **procedure** GETREPRESENTATIVEPOINTS($image, contour, num\_points$)
9:      Draw $contour$ on a mask
10:      Erode the mask
11:      Find points in the eroded mask
12:      **if** number of points $\leq num\_points$ **then**
13:          **return** points
14:      **else**
15:          Apply K-Means clustering to points to get $num\_points$
16:          **return** cluster centers as representative points
17:      **end if**
18: **end procedure**
19: **procedure** PROCESS($color\_name$)
20:      $contours, mask \leftarrow$ GETCOLORBOUNDARIES($image, color\_name$)
21:      Calculate total area of selected regions in $mask$
22:      **for** each $contour$ in $contours$ **do**
23:          Calculate $area\_ratio$ for the contour
24:          Determine $num\_points$ based on $area\_ratio$
25:          $contour\_points \leftarrow$ GETREPRESENTATIVEPOINTS($image, contour, num\_points$)
26:          Annotate $image$ with $contour\_points$
27:      **end for**
28:      **return** points
29: **end procedure**
30: **procedure** CONVERTPOINTSTOCOORDINATES($points$)
31:      Initialize lists for longitude $\lambda$ and latitude $\varphi$
32:      **for** each point $pt$ in $points$ **do**
33:          Calculate longitude and latitude based on $pt$ and image dimensions
34:          Append to $\lambda$ and $\varphi$ lists
35:      **end for**
36:      **return** $\varphi, \lambda$
37: **end procedure**
38: **procedure** GETCOR($color\_name$)
39:      $points \leftarrow$ PROCESS($color\_name$)
40:      $\varphi, \lambda \leftarrow$ CONVERTPOINTSTOCOORDINATES($points$)
41:      Print image dimensions
42:      **return** $\varphi, \lambda$
43: **end procedure**

---

## A.5 QUESTION AND ANSWER FORMAT TEMPLATE

Table 6: Template of Question and Answer Format

| Question Type | | Format |
|---|---|---|
| Verification | Question | Can you verify whether it is true or false that the location specified as **{Location Name}** is currently experiencing strong gales, storm or hurricane? |
| | Answer | True/False |
| Enumeration | Question | Could you identify the **{ "Land" or "Ocean"}** regions currently undergoing strong gales, storm or hurricane? |
| | Answer | **{Continent Name or Ocean Name}: {Province or State Name}** |
| Geo-Indexing | Question | What is the latitude and longitude of the location referred to as **{Location Name}**? |
| | Answer | **{Coordinate e.g.(-58.82, 176.31)}** |
| Description | Question | Can you describe the image in detail? |
| | Answer | Globally, regions are battling intense weather phenomena, including powerful gales, storms, and hurricanes, with wind gusts surpassing 20.7 m/s. The fury of these winds is felt over vast lands and oceanic stretches alike. Notably, areas such as **{Ocean Name in Red Color}**. Closer to human habitation, land regions in **{Land Name in Red Color}** reel under the power of these gales, showing nature's unbridled force across both developed and developing landscapes. Moreover, a vast array of regions encounters strong breezes and winds gusting between 10.8 to 20.7 m/s, impacting both land and sea. This includes **{Ocean Name in Yellow Color}**. On land, **{Land Name in Yellow Color}**, each face their own challenges with these forceful winds that spare few corners of the Earth. |

## A.6 GPT-4 EVALUATION PROMPT

Table 7: GPT-4 Evaluation Prompt

| | |
|---|---|
| **Prompt** | The user question is {question}. the ground truth answer is {gt_ans}, the generated response {gpt_ans} is generated by GPT model. |
| | Please act as an impartial judge and evaluate the quality of the response provided by AI assistant to the question displayed upper! You should give three scores to the response. the highest score is 5 and the lowest score is 1. the scores include: |
| | Score 1: the total score considering factors helpfulness, relevance, accuracy, depth, creativity, and level of detail of the generated response. |
| | Score 2: the similarity and completeness score between ground truth answer and generated response. sometimes the generated answer have mention some inaccurate point or the answer is incomplete compared with the ground truth answer. 5 means 80%-90% similar. 4 means 60%-80% similar. 3 means 40%-60% similar, 2 means 20%-40% similar and 1 means 0-20% similar. |
| | Score 3: the total score considering factors helpfulness, relevance, accuracy, depth, creativity, and level of detail of the ground truth answer. |
| | Avoid any position biases and ensure that the order in which the responses were presented does not influence your decision. Do not allow the length of the responses to influence your evaluation, Be as objective as possible. Directly output the score and strictly follow the format: |
| | ### Score 1: number ### Score 2: number ### Score 3: number |

