# OpenReview forum: "Vision-Language Models Meet Meteorology: Developing Models for Anomalies Analysis with Heatmaps"
_ICLR.cc/2025/Conference — ICLR 2025 Conference Withdrawn Submission_

### Official Review · Reviewer_1S7A · 2024-10-31

**Soundness:** 2
**Presentation:** 3
**Contribution:** 2
**Rating:** 3
**Confidence:** 3

**Summary:**

The paper presents evaluation of vision-language models (VLM) for meteorological anomaly detection and a custom dataset and model for training VLM for anomaly detection. The authors evaluate GPT-4-Vision on their dataset. They then create CLIMATEIQA dataset, containing 274K Q & A pairs on 8760 images. Finally, they fine-tune 3 three state-of-the-art VLMs on this dataset. Finetuning on CLIMATEIQA significantly improves the accuracy over the state-of-the-art models.

**Strengths:**

The paper touches upon an important topic of meteorological anomalies detection which has impact on various downstream applications. The author conduct a systematic assessment of general-purpose VLMs, which they evaluate using a custom dataset. Fine-tuning on the proposed dataset seems to significantly improve the performance of VLM on the the prescribed tasks.

**Weaknesses:**

I have several major concerns with the paper.

The approach relies on meteorological data presented in a RGB image of the world. I find it problematical to begin with. Instead of taking  input as geo-coded meteorological data. The presented approach converts it into a format which is designed for human-visualization. It then compares the performance of VLM on the converted meteorological data. I would argue that they are numerous ways in which meteorological data can be represented as images. For the example, most world maps are centered on the geographic lat, long of 0,0. But authors use a different perspective of 0,0 (lat, long) being on extreme left. While there is nothing wrong with the choice, the fact that it differs from how most world maps are visualized might be causing some of the misclassification shown in the examples (eg. Figure 2 "Across central Asia" misclassification). It is not clear to me whether the VLM are simply learning the perspective used by the authors? This bring me back to the point of presenting geo-coded meteorological data as images. I see several disadvantages in it, and more it makes is difficult to evaluate if the models lacking understanding of the geological perspective or something else.

Moreover, the paper starts with the argument that "Traditional methods rely on numerical threshold setting and manual interpretation of weather heatmaps with Geographic Information Systems (GIS), which can be slow and error-prone". Neither does the paper show any comparison with "traditional methods", nor does it offer any argument to why there method is faster. Unlike many of the traditional method, the data is projected onto a ill suited format, where the user has little control over the precise sampling resolution or the type of projection. The chosen projection i.e. Mercator is a conformal cylindrical map projection that was originally created to display accurate compass bearings for sea travel. It preserves angles at the cost of distorting areas and I don't see why image in Mercator projection is a good choice for training a VLM meant to predict extreme weather events over distorted landmass.

Finally, I fail to see how there approach is better than the numerical threshold setting, which they aim to improve upon. The proposed method SPOT is uses exactly the same approach approach they criticize to begin with. The paper seems like a solving a problem with a method ill suited for the problem. While author do present excellent results on their trained VLM models, the solution is not general for other representations e.g. variations in colors or projections of the displayed maps.

**Questions:**

How well does the VLM work for other representations of the weather data ( e.g. variations in colors or projections of the displayed maps)?
In what ways is the method better than traditional methods?

---

### Official Review · Reviewer_uvEx · 2024-11-01

**Soundness:** 2
**Presentation:** 2
**Contribution:** 1
**Rating:** 3
**Confidence:** 4

**Summary:**

This paper introduces a new benchmark dataset, meteorological VQA dataset on identifying the meteorological anomalies. The author also introduces the Climate-Zoo, a collection of meteorological VLMs built upon state-of-the-art VLMs.

**Strengths:**

1. The goal of this paper is clearly presented and easy to follow.
2. Proposed a method SPOT, which reaches 100% accuracy.
3. The dataset is derived from the ERA5, which is very popular and has high credibility.

**Weaknesses:**

1. The practical utility of the Climate Zoo remains uncertain. The success of LLMs is often attributed to their generalizability and zero-shot capabilities. While it is generally accepted that fine-tuning can improve model performance on specific tasks within a domain, the question remains whether Climate Zoo can effectively adapt to unseen distributions. This adaptability is particularly crucial in climate applications, where 1) temporal shifts are frequent, 2) higher accuracy is required.

2. Overall, the proposed benchmark dataset appears too simplistic. Despite showing different tasks, they are originated from color localization skills. It does not require the LLM to demonstrate reasoning abilities but instead focuses on testing object detection and color recognition capabilities. One can argue that LLM fails on such a simple task. However, there are other better alternatives than LLMs. Therefore, authors should answer why we must make LLM work well while other options are presented (especially with the presence of SPOT), especially considering the low LLM's efficiency in inference. In summary, the motivation for employing LLMs for such straightforward tasks is not well justified.

3. Related to point 2, The prompt design in Experiment 4 resembles classic object detection setups, RCNN and its deviates, which indicates classic methods may have a better accuracy.

4. Lack of in-depth discussion in ablation study. If having more data doesn't help, it is possible to that the data quality is not very good. that possibility is not being ruled out.

5. Other information from ERA5, such as air pressure and geopential, is not being used.

**Questions:**

1. What does prompt tuning in Figure 2 refer to? Prompt tuning typically describes a technique that uses soft prompts to optimize the loss function. However, based on the context, it seems you might not be using this method. Do you mean prompt optimization instead?

2. What does "list the color names in grids" mean in the context of Figure 2? It is unclear to me why listing out the color names (red, green, etc) can help for the zero-shot performance.

3. Does the resolution impact the final results? What is the image size used, and is there any preprocessing involved (e.g., down-sampling)?

3. How is training and test data split?

4. In Table 1, Yi-VL-6B shows only a slight improvement in BLEU when fine-tuned with LoRA compared to the baseline. However, Qwen-VL-Chat and Llava-v1.6-mistral-7b show much more significant improvements. I believe the author points out this phenomenon in the text, but is there a reason for this discrepancy? Does increasing the rank of the LORA help?

---

### Official Review · Reviewer_pjzp · 2024-11-01

**Soundness:** 2
**Presentation:** 3
**Contribution:** 3
**Rating:** 5
**Confidence:** 3

**Summary:**

This paper explores meteorological anomaly analysis (MAA) with the help of Vision-Language Models (VLMs). Specially, the authors introduce the first meteorological VQA dataset ClimateIQA, including 8,760 wind gust heatmaps and 254,040 question-answer pairs covering four question types, i.e., verification, enumeration, geo-indexing, description. The paper introduces Sparse Position and Outline Tracking (SPOT), which leverages K-means clustering to enhance spatial localization accuracy in heatmaps. In addition, the Climate-Zoo, a collection of meteorological VLMs trained using ClimateIQA, outperform state-of-the-art general VLMs in MAA.

**Strengths:**

- The paper is well-organized, presenting the study’s background, methodology, and findings logically.
- The preliminary experiments highlight the limitations of general VLMs, justifying the need for specialized models.
- The new ClimateIQA, tailored for meteorological anomaly analysis, is a new benchmark for meteorological applications.

**Weaknesses:**

- Inconsistent Statements: The paper introduces SPOT as a solution for inadequate color differentiation. However, in L260, SPOT is described as addressing the issue of irregular shapes and obtaining spatial location and shape information, leading to inconsistency regarding SPOT’s core purpose.

- Insufficient Method Comparison: The authors mention that traditional GIS methods are time-consuming and error-prone, they don’t provide a quantitate performance comparison with traditional methods. Additionally, although preliminary experiments evaluate GPT-4V’s performance, an exploration of few-shot settings could offer a more comprehensive comparison.

- Missing Processing Details: Details on how initial points for each color region are obtained remain unclear in the main text. I would recommend the authors to provide a process diagram for clarity. Furthermore, SPOT’s accuracy is frequently stated as “100%,” but it is unclear if this refers only to the representative point accuracy or includes the contour accuracy.

- Absence of Expert Study: While the proposed method shows good results, a user study or feedback loop from domain experts could provide valuable insights into practical usability and areas for further improvement.

Typos:
- L293: shwon => shown
- L317: List Questions => Enumeration Questions.

**Questions:**

Please refer to the Weaknesses.

---

### Official Review · Reviewer_yvY7 · 2024-11-04

**Soundness:** 3
**Presentation:** 4
**Contribution:** 3
**Rating:** 5
**Confidence:** 4

**Summary:**

This paper presents a systematic study of using vision-language models for Meteorological Anomalies Analysis by framing them as visual question answering problems. A large-scale dataset, ClimateIQA, was constructed which includes 8,760 wind gust heatmaps and 254,040 question-anwering pairs. By fine-tuning on this dataset, they present Climate-Zoo, the first meteorological VLM collection.

**Strengths:**

1. I personally enjoyed reading this paper since I loved seeing the VLM study on geoscience tasks. The authors conduct a systematic study on how VLM can be used for Meteorological Anomalies Analysis. The results are encouraging.
2. The paper is well-written and easy to follow.
3. The ClimateIQA dataset and Climate-Zoo model collection have potentials to be used for meteorological research.

**Weaknesses:**

1. One big concern is that the question and answer pairs are not written by humans but automatically generated with templates shown in Table 6. This will lead to low question diversity. I am worried that the reason why fine-tuned VLMs show very good performance is because they overfit the presented QA templates. Could you slightly change the QA template a bit and reevaluate the fine-tined VLMs to see how those models perform?

2. Other than using those large VLM as backbones, can you try other small VQA models as supervised baselines at least for verification questions?

3. I am not exactly sure why the representative point coordinates are needed. The paper does not provide a reason why we need this. Also why do you need to geo-locate it on the map? If the points are located on the ocean, how could you retrieve the location names?

4. What is the difference between locations and coordinates in Figure 3?

**Questions:**

See the weakness.

---

### Official Review · Reviewer_YzJP · 2024-11-10

**Soundness:** 3
**Presentation:** 3
**Contribution:** 3
**Rating:** 8
**Confidence:** 5

**Summary:**

The paper attempts to improve the VLMs’ performance in heat map-based Meteorological Anomalies Analysis. Authors first conduct initially analysis of the ability of LVMs in this task and then propose a method to obtain data from weather heatmaps. A VQA dataset is then collected for finetuning the LVMs and improving the performance.

**Strengths:**

1. The paper conducts an interesting attempt in linking weather data with LVMs.
2. The analysis and collected VQA dataset are valuable.
3. Experimental results demonstrate the effectiveness of the proposed pipeline.

**Weaknesses:**

1. The construction of the data is mainly from weather heatmaps. It would be interesting to explore a dataset from the raw data.
2. The openness, I have not found the link for code and data, which is essential to boost the research in the area.
3. Compared with the traditional threshold setting, LVM-based methods seem not accurate and can only present a rough region name for the extreme region. The applications scenarios of such method may be limited and it would be interesting to explore how to obtain more accurate weather-languge pairs.

**Questions:**

please refer to the weakness

---

### Note · Authors · 2024-12-13

I have read and agree with the venue's withdrawal policy on behalf of myself and my co-authors.